# Physical Activity and Related Factors in Pre-Adolescent Southern African Children of Diverse Population Groups

**DOI:** 10.3390/ijerph19169912

**Published:** 2022-08-11

**Authors:** Adeline Pretorius, Paola Wood, Piet Becker, Friede Wenhold

**Affiliations:** 1Department of Human Nutrition, Faculty of Health Sciences, University of Pretoria, Private Bag X323, Arcadia 0007, South Africa; 2Department of Consumer and Food Sciences, Faculty of Natural and Agricultural Sciences, University of Pretoria, Private Bag X20, Hatfield 0028, South Africa; 3Department of Physiology, Faculty of Health Sciences, University of Pretoria, Private Bag X323, Arcadia 0007, South Africa; 4Research Office, Faculty of Health Sciences, University of Pretoria, Private Bag X323, Arcadia 0007, South Africa

**Keywords:** physical activity, pre-adolescent, objective measurement, subjective measurement, body composition, obesity

## Abstract

Tailored obesity management includes understanding physical activity (PA) and its context, ideally in childhood before the onset of health risk. This cross-sectional study determined, by sex and population, the PA of Southern African pre-adolescent urban primary school children. PA was measured objectively (step count: pedometer) and subjectively (Physical Activity Questionnaire for Older Children [PAQ-C]), taking confounders (phenotype, school-built environment, and socio-economic environment) into account. Body composition was measured with multifrequency bioelectrical impedance analysis (Seca mBCA). PA was adjusted for phenotypic confounders (body size and composition) using multivariate regression. Sex and population differences in PA were determined with two-way ANOVA. Ninety-four healthy pre-adolescents (60% girls, 52% black) with a similar socio-economic status and access to PA participated. Amidst phenotypic differences, average steps/day in girls (10,212) was lower than in boys (11,433) (*p* = 0.029), and lower in black (9280) than in white (12,258) (*p* < 0.001) participants. PAQ-C scores (5-point rating) were lower for girls (2.63) than boys (2.92) (*p* < 0.001) but higher for black (2.89) than white (2.58) (*p* < 0.001) participants. Objective and subjective measurements were, however, not significantly (r = −0.02; *p* = 0.876) related and PAQ-C failed to identify reactive changes in the step count. Objectively measured PA of black participants and of girls was consistently lower than for white participants and boys. Target-group specific interventions should therefore be considered.

## 1. Introduction

Physical activity (PA) is a modifiable component of total energy expenditure that can greatly affect energy balance [1]. Over the past decades, there has been a decline in PA due to a behavioral shift from traditionally active to more sedentary lifestyles, often referred to as the “PA transition” [2]. The World Health Organization (WHO) classifies physical inactivity as the fourth largest cause of global mortality and a major determinant of noncommunicable diseases (NCDs) [3]. Globally, more than 80% of children and adolescents are inactive [4]. For children and adolescents, higher levels of PA are associated with a reduced risk of overweight/obesity and the consequent development of preventable diet- and lifestyle-related disease later in life [5]. Consequently, the WHO calls for urgent and concrete guidance to improve health and wellbeing for all [4]. In South Africa (SA), there is currently a concern for the health and wellbeing of children, in part due to physical inactivity and sedentary behavior. More than half of Southern African (South. Afr.) children are not meeting PA recommendations [6], and children from different population groups appear to have different PA levels [7,8,9]. Since obesity and NCDs may originate in childhood and present differently across populations [10], understanding of PA patterns in children from different population groups is important to guide effective population-specific PA interventions [5,6,7,11]. However, the PA of children is a complex behavior, and variations, especially between population groups, may be influenced by many interrelated environmental, biological, and social factors [1,12,13]. Several studies and systematic reviews [1,12,13] determined factors associated with the PA of young children. Sex has been identified as one of the primary factors, with age becoming increasingly important during adolescence [1,7,8,13]. Additionally, the environment in which people live, as well as their socio-economic status (SES), are considered interrelated factors affecting the PA of children [7,8,10,11,14]. Furthermore, being physically inactive may lead to a chain of circumstances. When overweight, children are less likely to engage in PA than peers with a healthy weight [7,15,16]. Although hindrances to PA among children with a higher body weight may be due to a lack of access to resources, social constraints, or low fitness levels, other aspects such as self-consciousness when being active, body dissatisfaction or having lower self-efficacy, and victimization and bullying may lead to isolated sedentary activities [7,15,16]. Despite being difficult to discern specific influences, it is important to consider these factors in interpreting PA in children of diverse population groups.

Accurate measurement of PA poses many challenges. Reference methods such as doubly-labelled water or indirect calorimetry are often unavailable, expensive, and impractical [17]. Consequently, cheaper unobtrusive measurements have become increasingly popular. Although these instruments cannot directly measure actual energy expended, they can be used to determine PA habits and trends. A variety of methods are available to capture activity, steps, or accelerations. However, PA behavior is complex and multi-faceted, with no recognized “gold standard” measurement technique [18]. Furthermore, general measurement limitations are often amplified in children due to intermittent activity patterns and changes during growth and development [17,18]. No single method can quantify all aspects of PA behavior and using multiple complementary methods combining objective and subjective approaches is recommended [17,18].

Although accelerometers are often used to assess PA in children, they are often out of the reach of low-resourced countries like South Africa. Pedometers are cost effective and objective alternative tools to track PA by registering the number of steps during walking or running [18,19]. This standardized steps/day unit of measurement allows for universal interpretation and cross-population comparisons. Pedometers measure vertical displacement only from ambulation; thus, some types of activities such as cycling and swimming or sedentary activities such as playing computer games are missed [17,19]. However, since walking is the most commonly reported PA, steps/day indices covering multiple days are considered an effective measure of habitual PA [20]. Along with a detailed measurement protocol [19,21,22], pedometer measurements are considered a reliable and accurate measure of PA in children and adolescents [19,23].

Subjective methods, an indirect method to assess PA, may fail to reflect PA accurately. Yet, they are valuable for monitoring determinants of activity levels and behaviors that cannot be detected with objective methods [17,18]. Typical subjective methods include physical activity questionnaires (PAQs), interviews, surveys, and activity diaries, each assessing different dimensions of PA and associated with their unique strengths and limitations [17,24]. Although the preferred method depends on the activity and the aim of the assessment, PAQs are the most widely used [17]. Several PAQs are available with varying degrees of reliability and validity [17]. The Physical Activity Questionnaire for Older Children (PAQ-C) has repeatedly been described as a reliable tool for children aged 6 to 12 years [24,25,26,27,28]. It entails a 7-day recall questionnaire developed as a guided, self-administered instrument to assess children’s habitual moderate- to vigorous-intensity PA (MVPA) in a school environment during the school term [27,29]. In South. Afr. settings, the PAC-C has been used successfully for studies in urban primary school learners [9,30] and has demonstrated good test–retest reliability and internal consistency, including internal consistency among ethnic groups (ICC = 0.96) [25].

Nested in an umbrella study, this study aimed to describe and compare the PA of black and white South. Afr. girls and boys aged 6 to 9 years, attending two primary schools in the City of Tshwane metropolitan area, Gauteng province in South Africa. The PA was determined using objective and subjective methods while considering PA-related confounding factors (school-built environment, the children’s socio-economic environment, and phenotypic characteristics).

## 2. Materials and Methods

### 2.1. Study Design, Population and Sampling

For this cross-sectional study, healthy pre-adolescent (6- to 9-year-old; age calculated from reported date of birth and date of assessment) black and white South. Afr. children were recruited from two primary schools. Schools were purposively selected based on their urban location within 2 km from each other, their similar school-built environment and access to PA opportunities at the school, and the perceived similarity (before the study) in the SES of the schools and the families who enroll their children at these schools. Based on a parental report, South. Afr. children included nationalities from all countries south of the equator, and population groups were categorized according to Statistics SA classification. Children who were injured, ill, taking chronic medication, or those who did not assent to partake in the study were excluded.

Participants were recruited via an invitation letter distributed to parents of all learners in grades one to three, and were typically aged 6 to 9 years. Parents who consented were contacted to confirm that the inclusion criteria were met and to arrange an appointment for measurements.

The statistical consideration for sample size calculation of the umbrella study [31], was applicable to this sub-study. To have 90% power, when using a two-sided two-group Student’s paired *t*-test at the 0.05 level of significance, recruitment needed to continue until 60 participants from each population group were enrolled in the study for a total sample size of 120 participants.

### 2.2. Data Collection and Management

Children spend a significant amount of time at school; consequently, the school and its neighboring environment are considered significant factors that may influence the PA behavior of young children [14,32]. Including two different schools in this study necessitated a comparison of PA opportunities at the schools and neighboring areas. Surveys of the school built environment (buildings, amenities, areas, and equipment), as well as the surrounding neighborhood and factors such as personal or road safety that may affect active commuting, are often used to describe access to PA facilities and opportunities that may enhance the PA of children [32,33]. A qualitative survey, the International Study of Childhood Obesity, Lifestyle and Environment [34] School Audit Tool, ISAT [35], was used in this study to assess and describe the school built environment and aspects in the neighborhood linked to PA.

Before assessing the children, consenting parents were requested to complete a sociodemographic questionnaire to identify the familial SES based on income category, housing conditions, household services (water, sanitation, and electricity supply), employment status, and level of education. Venues close to the school or on the school grounds were used for anthropometric and body composition measurements in the morning (6h30–7h30) before school. Before each measurement, the study purpose and measuring procedures were explained, and assent was obtained. Anthropometric measurements were taken according to the protocol of the United States Centers for Disease Control and Prevention [36]. Standing height (in cm, to the nearest 0.1 cm) was measured with the Seca 274 digital mobile stadiometer (Hamburg, Germany) [37]. Readings were transferred wirelessly to the multifrequency Seca mBCA 514 (Hamburg, Germany) [37], which was used to measure weight (in kg, to the nearest 100 g) and bioelectrical impedance analysis (BIA), to determine body composition. Data were transferred wirelessly from the Seca mBCA to a personal computer. The software in Seca mBCA 514 was used to calculate z-scores for weight-for-age (WFA), height-for-age (HFA) and body mass index-for-age (BMI-FA), based on the WHO growth reference data [38], and displayed in Microsoft Excel. The BMI-FA z-score was used to classify participants according to the WHO weight categories [38], as being healthy (−2 ≤ BMI-FA z-score ≤ 1), overweight (1 < BMI-FA z-score ≤ 2), or obese (BMI-FA z-score > 2) with BMI referring to weight in kilograms divided by height in meters squared.

The resistance values (R, measured at 50 Hz) were manually transferred to Microsoft Excel to calculate FFM (kg) using the equation of Horlick [39]. Fat mass (kg) was calculated by subtracting the FFM from total body weight (kg). Fat-free mass index (FFMI) and fat mass index (FMI) were, respectively, calculated by dividing FFM and FM (both in kg) by height in meters squared.

A full service of measurement instruments prior to the study and daily calibration was performed according to manufacturer instructions.

A spring-levered Yamax Digi-Walker SW-800 pedometer (Yamasa, Japan) was used for step counting. Due to the lack of a standardized protocol for pedometer measurements in young children, guidance from previous research [19,21,22], was used to develop measurement procedures: participants were requested to wear the pedometer for seven consecutive days from when they woke up in the morning until they went to bed at night [21]. Parents were requested to report any failure to adhere to this protocol, in which case participants were requested to wear the pedometer for an additional day. The device was securely fitted with an adjustable band around the waist to prevent tilting and consequent diminished sensitivity [19]. Reactivity and device tampering were prevented by covering the pedometer with masking tape each day [19,21,22,40]. Participants were randomly monitored after school to ensure wearing instructions were adhered to. Parents were requested to inform the investigator (AP) via text message of the step count reading at the end of each day. The daily step count readings were captured, and the average steps/day (across the seven days), steps/weekday, and steps/weekend day were calculated for each participant in Microsoft Excel.

Participants completed the PAQ-C before and after the seven days of wearing the pedometer, with the aim of monitoring device reactivity while wearing the pedometer. Although the PAQ-C was developed as a self-administered questionnaire, it has been suggested that the validity of self-administered methods in children can be increased by interviewer administration [18,29]. Therefore, the investigator assisted participants with completing the PAQ-C by reading the questions and available options before indicating their chosen option on the questionnaire. Guided by previous studies [9,28,30], the PAQ-C was translated into Afrikaans for participants from school A, an Afrikaans teaching medium school and the existing English PAQ-C was used at school B, an English teaching medium school. A slight adaptation was made to the list of activities to include the most popular local sports activities, e.g., in-line skating was replaced with ice/roller skating, basketball with netball, badminton with tennis, and street hockey with hockey. The adapted and translated questionnaires were tested during the pilot study. The PAQ-C questions were scored on a 5-point scale to evaluate the frequency and intensity of activities. A higher score indicated a higher level of activity. The mean of these items formed a final composite activity score between 1 and 5 [29]. The PAQ-C score was calculated as an average of the assessments before and after the pedometer assessment period.

### 2.3. Data Analysis

Stata Statistical Software Release 15 (StataCorp., 2017, StataCorp LCC, College Station, TX, USA) was used for statistical analysis. The significance level was set at 0.05 for all analyses. Differences in SES (categorical variables) between population groups and between schools were assessed using the two-sided Fisher’s exact test. Continuous variables were summarized by sex and population group, reporting linear-estimated means (predictive margins) including a 95% confidence interval, following an analysis of variance (ANOVA) with fixed effects sex, population group, and their interaction. In addition, two-way ANOVA was used to examine the effect of sex and population group and their interaction on the measured PA variables. Student’s paired *t*-test was used to assess the difference in the average steps/weekday and steps/weekend day, and between the mean PAQ-C-before and after wearing the pedometer. Spearman’s product–moment correlation, assuming a non-normal distribution, was used to determine the strength and statistical significance of the relationship between objectively and subjectively measured PA (average steps/day, and PAQ-C average) and between PA and each relevant variable including age, WFA z-score, HFA z-score, BMI-FA z-score, FFM, FFM, FM, and FMI, respectively. Although not a primary objective, the Welch two-sample *t*-test with unequal variances was used to explore and determine the difference in the average steps/day between population groups and between sexes per healthy and overweight/obese weight categories. Finally, the associations of sex and population group with respect to PA were assessed with multivariate regression while taking interactions of these same covariates into account.

## 3. Results

### 3.1. Outline of School-Built Environment, Socio-Economic Background, and Phenotype of the Sample

The sample of the umbrella study has been described previously [31]. In total, 94 participants (mean age 7.9 ± 0.79 years) were included. Fifty-six (60%) were girls (27 black; 29 white), and 49 (52%) were black.

The school-built environment, based on the ISAT results (qualitative data not shown), confirmed that both schools provided similar PA opportunities with access to good quality sports and recreational amenities. The school grounds provided a pleasing aesthetic environment, offering opportunities to interact with nature. Neither of the school’s surrounding neighborhoods provided optimal facilities for active transport to schools such as walking or cycling, whereas motorized transport was supported.

Socio-demographically, the schools and population groups were considered homogenous since income categories, home ownership and the number of household members, access to household services, and level of education did not differ significantly (all *p* > 0.05). Most households (90%) fell within the upper quintile income category, with multiple income sources and no unemployment. All caregivers completed secondary education, and 96% attained a tertiary qualification. Housing conditions (home ownership and the number of members per household) and access to household services (access to water, sanitation, and electricity supplies) reflected a higher standard of living.

In terms of phenotypic characteristics of the children, the following was found [31]: for both sexes, no significant differences were observed between z-scores of WFA, HFA, and BMI-FA (*p* = 0.997; 0.820; 0.795, respectively). However, the mean FFM and FFMI were significantly lower (1.73 kg and 0.93 kg/m^2^, respectively; *p* < 0.001 for both) for girls than boys. In contrast, the mean FM and FMI were significantly higher (1.71 kg; *p* = 0.035 and 1.08 kg/m^2^; *p* = 0.010 respectively) for girls than boys. Between population groups, the mean HFA z-score and FFM were significantly lower (0.97; *p* < 0.001 and 1.45 kg; *p* = 0.003, respectively) for the black than the white group. Conversely, the BMI-FA z-score, FM and FMI were respectively 0.46 (*p* = 0.042), 2.12 kg (*p* = 0.008), and 1.37 kg/m^2^ (*p* < 0.001) higher for the black than white group. No statistically significant differences were observed between the population groups for the WFA z-score and FFMI.

Most participants (83%), and all the white boys, had a healthy weight. More black than white children were within the overweight (14.3% and 4.5% respectively) or obese (12.2% and 2.2% respectively) category. Only 5% of boys, compared to 13% of the girls, were within the overweight category, with an almost equal distribution between girls (7.1%) and boys (7.9%) in the obese category.

### 3.2. Objective and Subjective PA

A summary of the measured PA variables for the sample is presented in Table 1. The average steps/weekday (10,996 ± 2927) was significantly higher (11%; *p* = 0.031) than the average for weekend days (9823 ± 4348).

The PAQ-C score before wearing the pedometer (2.67 ± 0.53) was statistically significantly lower (6%; *p* = 0.002) than the PAQ-C score after wearing the pedometer for one week (2.84 ± 0.44).

Table 2 summarizes the PA measurements and related variables by sex and population group. The means of all measured step count variables were significantly lower (*p* < 0.001) for black than white participants. Similarly, the mean values for girls were lower than for boys, although only the average steps/day and average steps/weekday (1220; *p* = 0.029 and 1177; *p* = 0.035 respectively) were considered statistically significantly lower for girls than for boys.

In contrast to the objectively measured PA, all PAQ-C scores were significantly higher (*p* < 0.05) for black than white participants. Additionally, and in agreement with objectively measured PA, all PAQ-C scores were significantly lower for girls than boys (*p* < 0.05).

No significant relationship between the means of the PAQ-C average score and the average steps/day for the total sample (r = −0.02; *p* = 0.876), nor for black participants (r = 0.15; *p* = 0.299), was seen (Table 3). For white participants a statistically significant, yet small correlation was observed (r = 0.37; *p* = 0.013) and for boys, this was a significant, negative relationship (r = −0.38; *p* = 0.020).

Based on the lack of correlation between the step count and the PAQ-C for the sample, and the assumption that the objectively measured average steps/day (step count) was a superior reflection of PA, the latter was used in the further analyses and interpretation of PA.

### 3.3. Factors Related to Step Count Measurements

Table 4 summarizes the relationship between average steps/day versus age and phenotypic variables, respectively.

For the total sample, a significant and positive relationship existed between steps/day and HFA z-score (r = 0.38; *p* < 0.001) and FFM (r = 0.36; *p* < 0.001), respectively. A similar relationship with HFA z-score (r = 0.50; *p* < 0.001) and FFM (r = 0.43; *p* = 0.007) was observed for boys. For black participants, a significant and positive relationship was also present between average steps/day and FFM. A significant and negative relationship existed between average steps/day and FMI (r = −0.26; *p* = 0.011) and in particular among girls.

Table 5 shows that similar to the results observed for the mean average steps/day (Table 2), the difference in average steps/day between population groups remained statistically significant (*p* < 0.001) after adjustment for age and phenotypic variables. However, between sex categories, the objective PA remained significantly lower (*p* < 0.05) for girls than for boys after PA was adjusted for age, FM and, anthropometric-related variables, including z-scores of WFA, HFA and, BMI-FA. When adjusted for FFM, FFMI and, FMI the sex differences decreased (*p* > 0.05).

Table 6 indicates the population differences in the mean average steps/day per weight category. Girls with obesity (BMI-FA z-score > 2) (n = 4) were grouped together with girls who were overweight (n = 7), due to the small number of participants in these weight categories.

Across weight categories, the mean average steps/day remained lower for black than white participants. Although no statistically significant difference was found between population groups in the overweight category (*p* = 0.103), the mean average steps/day was 2292 (24%) lower for black than for white girls. This may be considered clinically significant, especially when considering the similar yet significant difference of 2057 steps/day (*p* = 0.010) indicated between population groups in the healthy weight category. Additionally, the mean average steps/day for girls in the healthy weight category was 1371 higher than for the girls in the overweight category, but the difference was also not statistically significant (*p* = 0.122). However, due to the small number of girls who were overweight and obese (n = 11), these results may not be conclusive.

## 4. Discussion

Overweight and obesity is considered a worldwide epidemic, and there is a need to address lifestyle changes from an early age to prevent long term health consequences [5]. The WHO recommends approximately 60 min per day of MVPA to minimize the risk of NCDs [32]. However, across the world, children from different cultures, populations, and age groups appear to have different PA levels [7,8,9], and no single intervention approach may appropriately promote PA.

The main finding of this study indicated that the mean step count significantly differed between population and sex groups, with the step count of pre-adolescent black participants and girls being consistently lower than for white participants and boys; the PA of pre-adolescent children was significantly lower during weekends than during the week, while PAQ-C outcomes failed to identify possible reactive changes in step count. For a more in-depth interpretation, the objective and subjective PA measurement of the sample are discussed in relation to existing literature while taking potential confounding factors into consideration.

### 4.1. Objective PA

Many studies [17,20,41,42,43], have associated steps/day indices with health outcomes in children and indicated the need for population and country-specific guidelines. Although no single set of guidelines is available, international studies recommend a daily step count range between approximately 10,000–15,000 and 12,000–18,000 for girls and boys, respectively [20,43]. More specifically, the Canadian CANPLAY study, based on an extended database, suggests 11,000–12,000 and 13,000–15,000 steps/day for girls and boys, respectively, aged 6 to 11 years, to meet the equivalent of approximately 60 min per day of MVPA [41,42]. Given these recommendations, the mean step count in our sample barely met the minimum suggested range [41,42]. Insufficient PA levels were specifically observed for the black participants in our study. These results are similar to the few studies [6,7,30], that previously described PA patterns in South. Afr. children. Despite limited data, especially for objective PA measurements, there is sufficient indication that South. Afr. children are not meeting PA requirements [6,7]. Malan and Nolte [30], measured step count in white South. Afr. pre-adolescent children in a similar research setting and reported an insufficient step count for girls and boys (7988 and 10,504 respectively) attending school in an urban neighborhood. Two larger-scale studies also indicated that at least 50% of pre-adolescent South. Afr. children, especially girls and those living in a disadvantaged neighborhood, did not meet the recommended MVPA [34,44,45]. In addition, a study including older South. Afr. children (5–18 years) [7], reported insufficient PA, especially among girls and black population groups, similar to the findings of our study.

The results of our study further indicated that PA during weekends was significantly lower than during the week. This was observed across sex categories and population groups. The difference in step count between sex categories and population groups increased during weekends, indicating that participants who were generally less involved in PA (girls and black children) were even less active during weekends than those with a generally higher PA (boys and white children). These results are similar to the findings of the South. Afr. study by Malan and Nolte [30], indicating a lower step count during weekends than weekdays across girls and boys. Additional evidence exists that more active pre-adolescent children, especially boys, were more likely to maintain their PA during weekends, whereas for less active peers, PA during weekends declined [46,47].

Fu et al. [48] reported that children who met the recommended step count (based on Tudor-Locke et al. [20]) during weekdays were less likely (36%; *p* = 0.02) to be overweight or obese than those not meeting the recommendations. For those meeting the recommendations during both weekdays and weekends, the odds were even lower (67%; *p* = 0.01) compared to those not meeting the recommendations. Meeting weekend step count guidelines only was not associated with weight status. They concluded that a positive relationship exists between recommended step count accumulation and the healthy weight status of children, which may be used to guide public health recommendations for the prevention of NCDs.

### 4.2. Subjective PA

No single method can quantify PA accurately and the use of a combination of objective and subjective methods is therefore recommended [17,29]. One of the disadvantages of using pedometers to measure PA is reactive changes in habitual activity behavior [40]. To monitor the possibility of reactive changes in PA of participants in this study, the PAQ-C [29], was introduced as an alternative subjective measure to assess PA before and after wearing the pedometer. In addition, subjective measurement can generally be used to add further insight into PA-related behavior. However, the use of subjective methods, including PAQs, presents various limitations such as inability to recall activities, dependence on response rate, and failure to identify PA time and intensity [17,24]. In children, PAQs may hold the extra limitation that a child may be less able to recall their PA, possibly due to their intermittent and variable activity patterns that may be harder to remember, and their differences in cognitive and linguistic ability by age may also play a role [17,18]. Despite precautionary efforts to overcome these limitations, including the use of a population-appropriate and validated tool, interviewee assistance, and training of the research assistant to provide interviewee support, the correlation between PAQ-C and step count measurements failed to meet statistical significance in the total sample. The PAQ-C outcomes could therefore not be applied to identify possible reactive changes in PA or provide further insight into the PA of the sample.

It should be noted that for the white participants, a moderate correlation was observed between the step count and PAQ-C measurements. This corresponds with previous research performed at a school in white South. Afr. children of a similar age (7–9 years), reporting a moderate relationship (r = 0.49; *p* < 0.001) between steps/day (averaged over 7 days) and PAQ-C [30]. It is possible that misinterpretation of PAQ-C questions due to a language barrier may explain the lower correlation observed in the black African population of this sample. Although the English communication skills of black participants were considered sufficient, Sepedi or Setswana is the home languages spoken by the majority of black families in Gauteng [49]. Language barrier is considered a typical limitation of PA questionnaires [17], that may also have influenced the outcomes of this study.

### 4.3. Factors Related to the PA of Pre-Adolescent Children

#### 4.3.1. The Influence of Sex and Age on PA

The results of this study and the above discussion showed that across various studies, the PA of pre-adolescent girls is consistently lower than for boys. These results are supported by two systematic reviews [1,13], including international as well as Sub-Saharan research, reporting that sex differences persist in children of all ages. Although age has also been identified as a possible predictor of PA in children, both systematic reviews [1,13], concluded that the effect of age may only surface during the adolescent years, with older children being less engaged in PA. This may explain the results of our study. The lack of correlation observed between step count and age may be due to the pre-adolescent stage of our participants, along with the narrow age range.

#### 4.3.2. PA of Population Groups

An international review [11], concluded that although population group is often related to PA, results from various studies are conflicting. This may be explained by evidence that similar populations and ethnic groups tend to cluster in the same sociodemographic areas, and it may often not be possible to distinguish between the contributory role of SES and population group [7,11]. Previous South Afr. Studies [7,8], indicated that white children were more engaged in PA than black children and that a lower SES was associated with lower activity. However, these observations may have been influenced by cultural differences since, in South Africa, black populations often associate a higher body weight, especially among females, with health and wealth and therefore tend to be less engaged in PA [50]. Our study was designed to recruit participants from a similar SES background to minimize its contributory effect. Nevertheless, it should be acknowledged that it may be difficult to differentiate the contributory roles and the lower PA of black than white participants in this sample may have been influenced by cultural differences.

#### 4.3.3. The Relationship between Phenotype and PA

When considering the influence of phenotypic characteristics on the PA of the total sample, a significant positive correlation was observed between step count and HFA z-score and FFM, respectively, in boys, indicating that taller boys with more FFM have higher PA. Although a significant correlation existed between step count and HFA z-score, the step count for boys remained almost unchanged when adjusted for HFA z-score, showing that PA was not influenced by stature. In addition, the correlation observed between step count and FFM, especially in black boys, decreased when FFM was expressed per unit of height, i.e., FFMI. This observed correlation between PA and FFM could therefore be explained by an increased FFM related to a taller stature.

When adjusted for FFMI the difference in step count between sexes decreased considerably. These findings support the above explanation that the higher FFM associated with a taller stature may partially explain the observed sex differences in PA. However, this only affects sex categories. Between population groups, the difference in step count remained significant after being adjusted for FFMI.

Between step count and FMI, a small yet significant and negative correlation was observed for girls, indicating that their higher FMI was related to a lower average step count. These results are not unexpected since previous studies reported that a higher body weight of children was inversely related to PA levels [7,48,51]. A systematic review by Kelley et al. [52] concluded that regular PA significantly reduced the BMI-FA z-score of children (3% reduction; *p* = 0.02) and a positive association between BMI and PA can be explained by insufficient PA. It was therefore surprising that in our study, no significant correlation was observed between PA and WFA z-score and BMI-FA z-score, respectively. However, this lack of a significant correlation observed in our study is not absolute. Although BMI is widely applied in research to determine associations between overweight or obesity and PA, it has the limitation of not differentiating between FM and FFM. Evidence suggests [51,52], that regular PA may decrease FM, but at the same time, it could increase FFM. Since FFM per kg contributes more to body weight than FM, the effect of PA on body weight is not constant. A perceived relationship, or the lack thereof, between body weight/BMI and PA may therefore be affected by body composition. In addition, body composition between populations may vary, and it is therefore suggested to consider body composition rather than BMI-FA alone when investigating PA of different populations [51].

Our study was not designed to investigate the PA of weight categories; nevertheless, we deemed it necessary to explore this since inactivity may lead to overweight and obesity and the consequent risk of adverse health outcomes [2,3,4,5,53]. Additionally, although statistical significance was not met, a negative correlation was observed between the average steps/day and BMI-FA, FM, and FMI, respectively. The number of participants who were overweight and obese in our study was insufficient to draw reasonable conclusions, yet results suggested that these children were less active than those within the healthy weight category. It is, therefore, evident from the PA outcomes that the worldwide PA transition associated with increasing levels of overweight and obesity may not have escaped South. Afr. children. Since children with obesity are more likely to become obese during adulthood [54], appropriate intervention is essential.

It should be noted that when PA was adjusted for all available variables, the population differences remained, whereas sex differences diminished, especially after adjustment for stature. Differences in PA between population groups are therefore not related to differences in phenotypic characteristics. However, body size may potentially affect PA levels across population and sex groups. Girls typically have a higher FM and often become less active during adolescence. An increased FM and body size may result in low self-esteem and consequent lower engagement in PA [7,15,16], posing an even greater risk of becoming overweight.

The results of this study, therefore, emphasize the need for a population-specific PA component in interventions aimed at preventing overweight, obesity, and related NCDs in South. Afr. children, especially for black children and girls.

### 4.4. Strenghts and Limitations

Limitations to our study include the lack in statistical analysis to determine possible confounding effects of SES on PA, even though participants were recruited from a relatively homogenous SES. Parental self-report was used to determine underlying illness that may have influenced PA behavior. Cost limitations necessitated the use of BIA instead of a gold standard measure such as DEXA for body composition analysis, and an estimation equation was used to calculate FFM using BIA resistance values. There were no validated equations available to estimate FFM specifically for South. Afr. children of various population groups, and the equation of Horlick [39], was the only one suitable for both black African and white children. The PAQ-C was not translated into the home language of all the participants and failed to determine reactive changes in PA while wearing the pedometer. Despite precautionary efforts, the subjective scores could not be used to provide further insight into the PA-related behavior of the participants.

Nevertheless, the study included many strengths by providing a comprehensive description of a multitude of factors within the study framework related to the PA of pre-adolescent children. It contributes to the limited data indicating that South. Afr. children may not meet PA recommendations to maintain health and identifies the requirement for population-specific PA intervention and recommendations. The study further identifies the need for a validated and reliable PAQ to be used across different population groups.

## 5. Conclusions

The measurement of habitual PA among pre-adolescent children is complex. Objective step counts may not concur with subjectively assessed PA, contributing to the complexity of accurate PA measurements. There is a need to investigate the use of appropriate subjective methods to gain further insight into PA patterns and behavior in this context.

Nonetheless, overall, objective PA measurement in this study identified that participants did not meet PA recommendations for weight and NCD risk management. Population group differences were evident, and girls, due to body size, may face a greater risk of reduced PA levels. As part of a holistic approach, this justifies the development and implementation of tailored PA interventions for pre-adolescent urban children in South Africa, with a particular focus on the black girl.

## Figures and Tables

**Table 1 ijerph-19-09912-t001:** Objective and subjective PA of pre-adolescent children (N = 94).

		Mean	SD ^a^	Minimum	Maximum	Calculated Difference	*p*-Value ^b^
Objective PA	Average steps/weekday	10,996	2927	4687	18,030	1173 ^c^	0.031
	Average steps/weekend day	9823	4348	1757	21,381		
	Average steps/day	10,663	3027	4033	18,271		
Subjective PA	PAQ-C score ^d^ before	2.67	0.53	1.62	3.99	−0.17 ^e^	0.002
	PAQ-C score ^d^ after	2.84	0.44	1.47	4.23		
	PAQ-C average score ^d^	2.75	0.41	1.86	3.83		

PA Physical activity; PAQ-C Physical activity questionnaire for older children; ^a^ SD Standard deviation of the mean; ^b^ Student’s paired *t*-test; ^c^ Average steps/weekday—average steps/weekend day; ^d^ PAQ-C score “before” and “after” wearing the pedometer; PAQ-C average score of “before” and “after” scores; Possible score range: min = 1, max = 5; ^e^ PAQ-C score before—PAQ-C score after.

**Table 2 ijerph-19-09912-t002:** PA by sex and population group (N = 94) ^a^.

	Sex	Mean ^b^	95% CI ^c^	Sex Difference Boys-Girls	*p*-Value ^d^	Population Group	Mean ^b^	95% CI ^c^	Population Difference Black- White	*p*-Value ^d^
Average steps/ day	Girls	10,212	(9519; 10,906)	1220	0.029	Black	9280	(8538; 10,022)	−2979	<0.001
	Boys	11,433	(10,588; 12,277)			White	12,258	(11,483; 13,033)		
Average steps/ weekday	Girls	10,554	(9862; 11,246)	1177	0.035	Black	9770	(9030; 10,510)	−2631	<0.001
	Boys	11,730	(10,887; 12,573)			White	12,401	(11,628; 13,174)		
Average steps/ weekend day	Girls	9345	(8318; 10,372)	1346	0.105	Black	7986	(6876; 9096)	−3916	<0.001
	Boys	10,691	(9424; 11,958)			White	11,901	(10,754; 13,048)		
PAQ-C score before ^e^	Girls	2.54	(2.41; 2.57)	0.30	0.002	Black	2.85	(2.72; 2.98)	0.41	<0.001
	Boys	2.84	(2.69; 2.99)			White	2.44	(2.31; 2.58)		
PAQ-C score after ^e^	Girls	2.72	(2.61; 2.83)	0.28	<0.001	Black	2.93	(2.82; 3.05)	0.21	0.013
	Boys	3.00	(2.87; 3.13)			White	2.72	(2.60; 2.84)		
PAQ-C average Score ^e^	Girls	2.63	(2.53; 2.72)	0.29	<0.001	Black	2.89	(2.80; 2.99)	0.31	<0.001
	Boys	2.92	(2.81; 3.03)			White	2.58	(2.48; 2.69)		

PA Physical activity; PAQ-C Physical activity questionnaire for older children; ^a^ Girls n = 56; boys n = 38; black n = 49; white n = 45. ^b^ Adjusted mean: Predictive margins of the general linear model for ANOVA with the factors sex and population group and their interaction; ^c^ 95% Confidence interval around the mean; ^d^ Two-way ANOVA; ^e^ PAQ-C score “before” and “after” wearing the pedometer; Possible score: min 1, max 5.

**Table 3 ijerph-19-09912-t003:** Correlation (r) between average steps/day and PAQ-C-average by sex and population group (N = 94).

Sex	r ^a^	*p*-Value ^b^	Population Group	r ^a^	*p*-Value ^b^	Total Sample	
						r ^a^	*p*-Value ^b^
Girls (n = 56)	0.20	0.133	Black (n = 49)	0.15	0.299	−0.02	0.876
Boys (n = 38)	−0.38	0.020	White (n = 45)	0.37	0.013		

PAQ-C Physical activity questionnaire for older children; ^a^ Spearman’s product-moment correlation; ^b^ Level of statistical significance.

**Table 4 ijerph-19-09912-t004:** Relationship between average steps/day versus age and phenotypic factors by sex and population group (N = 94) ^a^.

Variable	Sex	r ^b^	*p*-Value ^c^	Population Group	r ^b^	*p*-Value ^c^	Total Sample	
							r ^b^	*p*-Value ^c^
Age	Girls	−0.04	0.745	Black	0.17	0.250	−0.02	0.875
	Boys	0.02	0.895	White	0.04	0.770		
WFA z-score	Girls	0.03	0.826	Black	0.05	0.718	0.09	0.381
	Boys	0.15	0.361	White	0.02	0.877		
HFA z-score	Girls	0.26	0.053	Black	0.25	0.081	0.38	<0.001
	Boys	0.50	<0.001	White	0.06	0.688		
BMI-FA z-score	Girls	−0.12	0.373	Black	−0.06	0.688	−0.14	0.185
	Boys	−0.16	0.347	White	−0.02	0.875		
FFM (kg)	Girls	0.24	0.080	Black	0.35	0.013	0.36	<0.001
	Boys	0.43	0.007	White	0.20	0.188		
FFMI (kg/m^2^)	Girls	0.13	0.341	Black	0.14	0.350	0.19	0.069
	Boys	0.14	0.400	White	0.23	0.129		
FM (kg)	Girls	−0.22	0.108	Black	−0.04	0.792	−0.20	0.056
	Boys	−0.143	0.393	White	−0.24	0.117		
FMI (kg/m^2^)	Girls	−0.27	0.043	Black	−0.10	0.490	−0.26	0.011
	Boys	−0.21	0.207	White	−0.24	0.109		

WFA Weight-for-age; HFA Height-for-age; BMI-FA Body mass index-for-age; FFM Fat-free mass; FFMI Fat-free mass index; FM Fat mass; FMI Fat mass index; ^a^ Girls n = 56; boys n = 38; black n = 49; white n = 45; ^b^ Spearman’s product-moment correlation; ^c^ Level of statistical significance.

**Table 5 ijerph-19-09912-t005:** Average steps/day adjusted for covariates by sex and population group (N = 94) ^a^.

Adjusted for ^b^	Sex	Average Steps/Day ^c^	95% CI ^d^	Sex Difference Boys-Girls	*p*-Value ^e^	Population Group	Average Steps/Day ^c^	95% CI ^d^	Population Difference Black-White	*p*-Value ^e^
Age	Girls	10,202	(9509; 10,896)	1239	0.027	Black	9217	(8464; 9970)	−3105	<0.001
	Boys	11,441	(10,596; 12,286)			White	12,322	(11,536; 13,108)		
WFA z-score	Girls	10,213	(9516; 10,909)	1220	0.030	Black	9293	(8545; 10,041)	−2951	<0.001
	Boys	11,433	(10,584; 12,228)			White	12,244	(11,463; 13,026)		
HFA z-score	Girls	10,217	(9530; 10,905)	1200	0.031	Black	9519	(8724; 10,314)	−2472	<0.001
	Boys	11,417	(10,579; 12,255)			White	11,991	(11,152; 12,830)		
BMI-FA z-score	Girls	10,213	(9516; 10,910)	1217	0.030	Black	9293	(8539; 10,047)	−2950	<0.001
	Boys	11,430	(10,580; 12,279)			White	12,243	(11,453; 13,032)		
FFM (kg)	Girls	10,391	(9692; 11,090)	775	0.183	Black	9457	(8711; 10,202)	−2606	<0.001
	Boys	11,166	(10,302; 12,030)			White	12,062	(11,281; 12,843)		
FFMI (kg/m^2^)	Girls	10,339	(9595; 11,083)	915	0.156	Black	9319	(8572; 10,066)	−2904	<0.001
	Boys	11,254	(10,328; 12,180)			White	12,223	(11,443; 13,002)		
FM (kg)	Girls	10,233	(9529; 10,937)	1169	0.042	Black	9310	(8551; 10,070)	−2915	<0.001
	Boys	11,402	(10,540; 12,263)			White	12,225	(11,429; 13,020)		
FMI (kg/m^2^)	Girls	10,259	(9554; 10,965)	1102	0.057	Black	9351	(8585; 10,117)	−2828	<0.001
	Boys	11,362	(10,495; 12,228)			White	12,179	(11,377; 12,982)		

WFA Weight-for-age; HFA Height-for-age; BMI-FA Body mass index-for-age; FFM Fat-free mass; FFMI Fat-free mass index; FM Fat mass; FMI Fat mass index; ^a^ Girls n = 56; boys n = 38; black n = 49; white n = 45; ^b^ Multivariate regression analysis; ^c^ Adjusted mean: predictive margins of the general linear model for ANOVA with the factors sex and population group and their interaction; ^d^ 95% Confidence interval around the mean; ^e^ Two-way ANOVA.

**Table 6 ijerph-19-09912-t006:** Mean daily step count per weight category by sex and population group (N = 94) ^a^.

Weight Category	Population Group	Girls				Boys			
		Average Steps/Day	SD ^b^	Population Difference Black-White	*p*-Value ^c^	Average Steps/Day	SD ^b^	Population Difference Black-White	*p*-Value ^c^
Healthy ^d^ (n = 78)	Black	9381	2375	−2057	0.010	9059	2831	−4496	<0.001
	White	11,437	2621			13,555	2498		
	Total	10,569	2696			11,239	3484		
Overweight ^e^/Obese ^f^ (n = 16)	Black	8573	1673	−2292	0.103				
	White	10,865	2424			No values ^g^			
	Total	9198	2069						

^a^ Girls n = 56; boys n = 38; black n = 49; white n = 45. ^b^ SD Standard deviation of the mean. ^c^ Welch two-sample *t*-test with unequal variances. WHO weight categories: Body mass index for-age z-score (BMI-FA z-score). ^d^ Healthy: −2 ≤ BMI-FA z-score ≤ 1 (No child with BMI-FA-z < −2). ^e^ Overweight: 1 < BMI-FA z-score ≤ 2. ^f^ Obese: BMI-FA z-score > 2. ^g^ No *p*-values since no white boys within the overweight/obese category.

## Data Availability

The data presented in this study are available on request from the corresponding author.

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
