# Peer review of "Physical Activity and Related Factors in Pre-Adolescent Southern African Children of Diverse Population Groups"

_ijerph, 2022, doi:10.3390/ijerph19169912_

Round 1

Reviewer 1 Report

Dear Authors,

thanks for the possibility to read such complex and detailed article!

My only recomendation is, that you should clearly articulate why your research is big, significant and what new it brings to the knowledge in the topic. According to this some parts of introduction and discussion/conclussion shoul be rebuild/extended.

Author Response

Thank you for the comments and suggestions.

We addressed this by clarifying the significance of this research and adding further recommendations by adjusting lines:

373-378; 386-389; 607-624; 644-653.

Reviewer 2 Report

Title

Overall, the manuscript title reads well and might spot interest in the reader.

Abstract

Overall, the abstract is well written. I have only some concerns. Firstly, Authors could provide some more specific information about the study sample. Secondly, Authors could specify the meaning on values in the brackets. Thirdly, Authors could provide some most important conclusions.

Introduction

Authors provide a great overview about the physical activity related research. Authors argue that accelerometers were not used in this study because these don’t reach to the low-resourced countries. I agree with the Authors that this is the issue. However, there are several great questionnaires that enable to gain deeper insight in this field of research, and they are very cost-effective. Moreover, Authors argue that they seek to understand physical activity and its context. For example, research in the field of self-determination theory (Ryan & Deci, 2000) has demonstrated that different forms of motivation are related to physical activity among youngsters. Specifically, recent research has shown that intrinsic motivation is the greatest predictor of daily physical activity among adolescents (Kalajas-Tilga et al., 2020). In turn, intrinsic motivation can be supported by providing youngsters autonomy, competence, and relatedness support.

Kalajas-Tilga, H., Koka, A., Hein, V., Tilga, H., & Raudsepp, L. (2020). Motivational processes in physical education and objectively measured physical activity among adolescents. Journal of Sport and Health Science, 9(5), 462–471. https://doi.org/10.1016/j.jshs.2019.06.001

Ryan, R. M, & Deci, E. L. (2000). Intrinsic and Extrinsic Motivations: Classic Definitions and New Directions. Contemporary Educational Psychology. 25(1), 54–67. https://doi.org/10.1006/ceps.1999.1020

Method

Overall, the method is clear. However, Authors could provide more specific information about study participants and data collection.

Results

Results are clear and accurate.

Discussion

Currently, the discussion is too much results oriented. Authors should avoid the repetition of the results in the discussion section. Authors could end the discussion with clear practical recommendations for the reader. Also, the conclusions sections should be more specific about the main results of the current study.

Reviewer 3 Report

Data Analysis and Results:

- ANOVA - inform about requirements' tests and estimate and present effect size and test power

- Student test - identify, estimate and present effcet size

- Correlation - estimate and present CI (confidence interval), preserving signs (Tables 3 & 4)

Tables 3 and 4 - remove bold in the the second line

Author Response

Thank you for the recommendations.

The comments were addressed as indicated below:

1. Adjusted line 228 - 233: “The significance level was set at 0.05 for all analysis. Continuous variables were summarized by sex and population group, reporting linear estimated means (predictive margins) including a 95% confidence interval, following an analysis of variance (ANOVA) with fixed effects sex, population group and their interaction. In addition, two-way ANOVA was used to examine the effect of sex and population group and their interaction on the measured PA variables.”

2. The test power was indicated in line 146: “To have 90% power, when using a two-sided two-group Student’s paired t-test at the 0.05 level of significance.”

3. Tables 3 and 4:

The formatting was addressed and bold font removed in tables 3 and 4.

The relevant mean values and 95% CI are presented in table 2.

Round 2

Reviewer 2 Report

Authors have done well job on revising their manuscript.

Author Response

Thank you for the feedback. The spelling and grammar errors have been addressed throughout the document.